# Population Structures and Dynamics of *Rhododendron* Communities with Different Stages of Succession in Northwest Guizhou, China

**DOI:** 10.3390/plants13070946

**Published:** 2024-03-25

**Authors:** Yaoyao Zhang, Jianli Wang, Xiaojing Wang, Lingjun Wang, Yuefeng Wang, Junpeng Wei, Zijing Niu, Linye Jian, Baocheng Jin, Chao Chen, Xuechun Zhao

**Affiliations:** 1College of Animal Science, Guizhou University, Guiyang 550025, China; gs.yaoyaozhang21@gzu.edu.cn (Y.Z.); wjl_kangta@163.com (J.W.); w18722932238@163.com (Y.W.); gs.jpwei22@gzu.edu.cn (J.W.); gs.zjniu23@gzu.edu.cn (Z.N.); gs.lyjian23@gzu.edu.cn (L.J.); bcjin@gzu.edu.cn (B.J.); chenc@gzu.edu.cn (C.C.); 2College of Data Science and Information Engineering, Guizhou Minzu University, Guiyang 550025, China; 3Institute of Agro-Bioengineering, Guizhou University, Guiyang 550025, China; xjwang8@gzu.edu.cn; 4Institute of Azalea, Baili Azalea Management Committee, Bijie 551614, China; 19311970957@163.com

**Keywords:** *Rhododendron* shrub, *Rhododendron annae*, *Rhododendron irroratum*, population structure, population dynamics

## Abstract

To explore the population structures and dynamics of *Rhododendron* shrub communities at different stages of succession in northwest Guizhou, China, this study examined the populations of *Rhododendron annae* and *Rhododendron irroratum* shrub with two different stages. A space-for-time substitution was employed to establish the diameter class/height structures, static life tables, and survival/mortality rate/disappearance rate curves of both *Rhododendron* populations with different orders of succession. Their structural and quantitative dynamics were analyzed, and their development trends were predicted. The results showed that, quantitatively, the populations of *R. annae* and *R. irroratum* in the two *Rhododendron* communities with different orders of succession were dominated by age classes one, two, and three as well as height classes i, ii, and iii. The number of *Rhododendron* plants at the three age classes and the three height classes accounted for 97.61–100% of the total. The quantitative dynamic indices of *R. annae* and *R. irroratum* were both greater than 0, with and without considering external interference. In terms of age class and height structures, both *Rhododendron* populations were expanding populations, presenting “inverted-J-shaped” and irregular pyramid patterns. There was a sufficient number of young individuals, but few or no old individuals. Both survival curves of the populations of *R. annae* and *R. irroratum* in the two *Rhododendron* communities with different orders of succession belonged to the Deevy-II type. In the late stage of succession, the mortality curves and disappearance curves of both *Rhododendron* populations in these communities presented a trend of increasing first and then decreasing with increasing age class. This result indicates that at each age class, *R. annae* and *R. irroratum* showed a trend of gradual increase after two, four, and six years. In brief, the populations of *R. annae* and *R. irroratum* have rich reserves of seedlings and saplings, but high mortality and disappearance rates. In this context, it is necessary to reduce human interference and implement targeted conservation measures to promote the natural renewal of *Rhododendron* populations.

## 1. Introduction

A population is the sum of individuals of the same species that occupy a certain space within a certain period. It reflects a link among individuals, communities, and ecosystems, and a fundamental component of communities [1]. The biological and ecological characteristics of plant populations are the result of long-term adaptation to and selection of environmental conditions. The dynamic changes in the size or quantity of a population on the spatiotemporal scale reflect the composition and development trends of the population; they also reflect the interactive relationship between the population and the environment and its position and role in the community [2,3]. The dynamics of population structure and quantity, which are among the core contents of ecological research, cover diameter class, height class, age class, and other items. Changes in these items directly affect community characteristics and objectively characterize the development and evolution trends of communities [1,4]. By examining the structure of a population using methods such as static life tables, survival curves, and survival analyses, it is possible to disclose the survival status of the population and the fitness and interactions of plants with the environment in the community [5,6,7]. Succession—the process of one community being replaced by another—is the most important characteristic of the dynamics of plant communities, an important pathway for ecological restoration and reconstruction, and the foundation for maintaining community dynamics stability and sustainable development [8,9]. In this sense, analyzing the structures and dynamics of plant populations is of great importance for elucidating the succession routes of plant communities.

*Rhododendron* (Ericaceae) is one of the largest genera of angiosperms, with important ornamental, cultural, scientific, economic, and medicinal values. There are over 1200 species of *Rhododendron* worldwide, over 900 of which are distributed in Asia and more than half of them in southwest China, which is the distribution and evolution center of *Rhododendron* plants in modern times [10,11,12,13,14]. Guizhou is located at the edge of the center and the transition zone of its eastward spread and is home to more than 110 naturally distributed *Rhododendron* species, second only to Yunnan, Sichuan, and Tibet [15,16]. Northwest Guizhou is the most important region in the distribution of the *Rhododendron* taxa in Guizhou, with a total of six subgenera and more than 50 species, accounting for about half of the *Rhododendron* plants in Guizhou. In this regard, the Baili Azalea Nature Reserve, located at the junction of Qianxi County and Dafang County, is the most representative and important habitat [17,18]. It is also a typical representative of the world’s largest and continuously distributed natural wild *Rhododendron* communities in middle-to-low-altitude mountainous areas [19]. In this region, *Rhododendron* shrub-based communities with different orders of succession cover grass rangelands, rose shrubs, *Rhododendron* shrubs, oak shrubs, and oak evergreen forests. However, the position, competitiveness, and stability of *Rhododendron* shrubs in these communities remain unclear.

Many scholars have conducted preliminary research and analyses on the *Rhododendron* shrubs in northwest Guizhou, China, from the perspectives of germplasm resources, community characteristics, and interference effects of *Rhododendron* plants [20,21,22]. However, these studies rarely touch upon the population structures or dynamics of *Rhododendron* in natural *Rhododendron* shrub communities at different stages of succession. This study takes the successional communities of *Rhododendron annae* and *Rhododendron irroratum*—two *Rhododendron* populations widely distributed throughout northwest Guizhou—as research objects. By analyzing the population structures and dynamics of *R. annae* and *R. irroratum* in two *Rhododendron* communities with different orders of succession, a theoretical basis is provided for the scientific protection, tourism development, and sustainable development of *Rhododendron* plants in northwest Guizhou.

The present study was conducted to achieve the following objectives: (1) to characterize the population structures or dynamics of *Rhododendron* in natural *Rhododendron* shrub communities at different stages of succession, (2) to explore the growth status of two *Rhododendron* species in communities at different successional stages, (3) to assess the self-renewal ability of the *Rhododendron* populations under natural conditions. 

## 2. Results

### 2.1. Quantification of Densities, Diameter Class/Height Structures, and Dynamics of Both Rhododendron Populations

The results of this survey found that a total of 765 *Rhododendron* plants were examined for the two orders of succession. To be specific, there were 430 *R. annae* plants (99, 186, and 145 in communities I_a_, II_a_, and III_a_, respectively) and 335 *R. irroratum* plants (73, 172, and 90 in communities I_b_, II_b_, and III_b_, respectively). The quantity of *R. annae* was 128.36% of that of *R. irroratum* (Figure 1).

*R. annae* (communities I_a_, II_a_, and III_a_) and *R. irroratum* (communities I_b_, II_b_, and III_b_) were both dominated by age classes one, two, and three. The total quantities of *R. annae* and *R. irroratum* for these three age classes accounted for 96.98% and 100.00% of the total quantity of *Rhododendron* plants, respectively. There were few *Rhododendron* plants in age class 4, 5, or 6, except for 12 *R. annae* plants. In terms of age structure, the two *Rhododendron* populations in each stage of succession were both expanding populations (Figure 2).

The dynamic changes in quantity between different age groups of *R. annae* and *R. irroratum* populations between age classes at different stages of succession are presented. In community I_a_, the quantitative dynamic indices of *R. annae* at age classes one and five (*V*_1_ and *V*_5_) indicated an increase in population size, whereas the quantitative dynamic indices at other age classes (*V*_2_, *V*_3_, *V*_4_, and *V*_6_) indicated a decrease in population size. In community II_a_, the quantitative dynamic indices of *R. annae* at all age classes (except age class 5; i.e., *V*_1_, *V*_2_, *V*_3_, *V*_4_, and *V*_6_) were greater than 0. Notably, *V*_5_ was equal to 0. In community III_a_, only the quantitative dynamic index of *R. annae* at age class 1 (*V*_1_) was less than 0, whereas those at age classes two and three (i.e., *V*_2_ and *V*_3_) were both greater than 0. The quantitative dynamic indices of *R. annae* with and without considering external interference (*V_pi_* and *V′_pi_*) were both greater than 0, with variation ranges of 39.57–48.15% and 2.67–7.90%, respectively. In communities I_b_, II_b_, and III_b_, the quantitative dynamic index of *R. irroratum* at age class one (*V*_1_) was uniformly less than 0, whereas those at age classes two and three (i.e., *V*_2_ and *V*_3_) were both greater than 0. The quantitative dynamic indices of *R. irroratum* with and without considering external interference (*V_pi_* and *V′_pi_*) were both greater than 0 as well, with variation ranges of 46.24–67.02% and 0.76–1.28%, respectively (Table 1).

*R. annae* (communities I_a_, II_a_, and III_a_) and *R. irroratum* (communities I_b_, II_b_, and III_b_) were both dominated by height classes i, ii, and iii. The total quantities of *R. annae* and *R. irroratum* at these three height classes accounted for 99.53% and 97.61% of the total quantity of *Rhododendron* plants, respectively. There were few *Rhododendron* plants at height class iv, v, or vi, except for twelve *R. annae* plants and eight *R. irroratum* plants. The individual numbers of *R. annae* (communities I_a_, II_a_, and III_a_) and *R. irroratum* (communities I_b_, II_b_, and III_b_) both presented a trend of increasing first and then decreasing with increasing height. The plant numbers of *R. annae* at height classes i and ii presented a trend of increasing first and then decreasing with orders of succession. There were 19, 23, and 6 *R. annae* plants at height class i, as well as 74, 114, and 81 *R. annae* plants at height class ii, respectively. The plant number of *R. annae* at height class iii exhibited a gradually increasing trend with orders of succession. There were 5, 114, and 81 *R. annae* plants at height class iii, respectively. At height classes iv and vi, communities III_a_ and I_a_ each contained one *R. annae* plant. The height structures of *R. annae* in the three communities with different orders of succession both represented the expanding type (Figure 3).

In communities I_b_, II_b_, and III_b_, the plant numbers of *R. irroratum* at height class i presented a trend of decreasing first and then increasing with orders of succession. There were eight, one, and ten *R. irroratum* plants at height class i, respectively. The plant numbers of *R. irroratum* at height classes ii and iii presented a trend of increasing first and then decreasing with orders of succession. There were 54, 77, and 25 *R. annae* plants at height class ii, as well as 11, 94, and 47 *R. annae* plants at height class iii, respectively. At height class iv, there were only eight *R. irroratum* plants in the community I_b_. The height structures of *R. irroratum* in the three communities with different orders of succession were also both of the expanding type (Figure 3b).

### 2.2. Static Life Tables and Survival Rate Curves of Both Rhododendron Populations in Communities at Different Stages of Succession

There were many young individuals (age classes 1–2) of *R. annae* in communities I_a_, II_a_, and III_a_. The survival numbers (*a_x_*) of *R. annae* in these three communities presented a trend of increasing first and then decreasing with increasing age class and all peaked at age class two, reaching 46, 105, and 93, respectively. The life expectancies (*e_x_*) of *R. annae* in communities I_a_ and II_a_ showed a trend of decreasing first, then increasing, and finally decreasing again with increasing age class, and peaked at age class four (I_a_) and age class one (II_a_), reaching 2.50 and 1.60, respectively. The life expectancy (*e_x_*) of *R. annae* in community III_a_ displayed a trend of gradual decrease with increasing age class, dropping from 1.35 at age class one to 0 at age classes four, five, and six. There were also many young individuals (age classes 1–2) of *R. irroratum* in communities I_b_, II_b_, and III_b_. With increasing age class, the survival numbers (*a_x_*) of *R. irroratum* in the three communities presented the same trend as those of *R. annae*, and also peaked at age class two, reaching 61, 135, and 57, respectively. The life expectancy (*e_x_*) of *R. irroratum* in community I_b_ showed a trend of decreasing first, then increasing, and finally decreasing again with increasing age class, and peaked at age class four, reaching 2.50. The life expectancies (*e_x_*) of *R. irroratum* in communities II_b_ and III_b_ displayed a trend of gradual decrease with increasing age class, dropping from 1.24 and 1.50 at age class one to 0 at age classes four, five, and six (Table 2). 

The survival, mortality, and disappearance curves of *R. annae* in communities with different orders of succession are shown in Figure 4. The logarithmic standardized survival numbers (*lnlx*) of *R. annae* in communities I_a_, II_a_, and III_a_ gradually decreased with increasing age class, and its survival curves all fell between Deevey-II and Deevey-III. The survival curves of *R. annae* in communities I_a_, II_a_, and III_a_ were tested using both exponential and power function models. As shown in Table 3, the *R*^2^ of the exponential function model was uniformly greater than that of the power function model; therefore, the survival curves of *R. annae* in communities I_a_, II_a_, and III_a_ tended to be closer to Deevey-II. The mortality rate and disappearance rate curves of *R. annae* in communities I_a_, II_a_, and III_a_ presented a consistent trend with increasing age class (Figure 4b,c). Specifically, both the mortality rate and disappearance rate curves of *R. annae* in communities I_a_ and II_a_ showed a trend of increasing first, then decreasing, and finally increasing again with increasing age class, and peaked at age classes three and six. The mortality rate and disappearance rate curves of *R. annae* in community III_a_ exhibited a trend of increasing first and then decreasing and peaked at age class three.

The logarithmic standardized survival numbers (lnl*x*) of *R. irroratum* in communities I_b_, II_b_, and III_b_ also gradually decreased with increasing age class, and all survival curves fell between Deevey-II and Deevey-III. As shown by the model tests in Table 3, the *R*^2^ of the exponential function model for *R. irroratum* in communities I_b_, II_b_, and III_b_ was uniformly greater than that of the power function model; therefore, the survival curves of *R. irroratum* in communities I_b_, II_b_, and III_b_ tended to be closer to Deevey-II. The mortality rate and disappearance rate curves of *R. irroratum* in communities I_b_, II_b_, and III_b_ both showed a trend of increasing first, then decreasing, and finally increasing again with increasing age class, and peaked at age classes three or four.

### 2.3. Time Sequence Analysis of the Two Rhododendron Populations in Communities at Different Stages of Succession

According to the population dynamics prediction of *R. annae* (Table 4), the population sizes of *R. annae* in communities I_a_, II_a_, and III_a_ will decrease from their current values of 99, 186, and 145 plants to 15, 27, and 14 plants in six years. The quantities of *R. annae* in communities I_a_, II_a_, and III_a_ all showed a trend of gradual increase after two, four, and six years at each age class, except for age class ii. The quantities at lower age classes always exceeded those at higher age classes. Specifically, the quantity of *R. annae* at age class two decreased by 39.13%, 40.00%, and 23.66%, respectively; that of *R. annae* at age class three increased by 17.65%, 44.64%, and 1500.00%, respectively. The quantities of *R. annae* in age classes four, five, and six also exhibited a gradually increasing trend.

According to the population dynamics prediction of *R. irroratum* (Table 4), the population sizes of *R. irroratum* in communities I_b_, II_b_, and III_b_ will decrease from the current numbers of 126, 172, and 90 plants to 12, 21, and 10 plants in six years. Similarly, the quantities of *R. irroratum* in communities I_b_, II_b_, and III_b_ all showed a trend of gradual increase after two, four, and six years at each age class, except for age class ii. Specifically, the quantity of *R. irroratum* at age class two decreased by 6.56%, 40.74%, and 31.58%, respectively, whereas that of *R. irroratum* at age class three increased by 227.27%, 469.23%, and 191.67%, respectively. The quantities of *R. irroratum* in age classes four, five, and six also exhibited a gradually increasing trend.

## 3. Discussion

### 3.1. Population Structures and Types

Population structures and dynamic features provide an important foundation for a better understanding of the survival status and dynamic development laws of populations [23]. The age class and height structures of a population can not only disclose the developmental stage of its individuals, but also elucidate the interactions between the biological characteristics of a species and its living environment [24]. The findings of this study indicate that the individual numbers of both *Rhododendron* communities with different orders of succession presented a trend of increasing first and then decreasing with the direction of succession. That is, communities II_a_ and II_b_ had the highest number of individuals (186 and 172, respectively). This is due to differences in competitiveness and status of the two *Rhododendron* populations in communities with different orders of succession. Both populations had large individual numbers in age classes 1–2, but the number of individuals at age class one was far lower than that at age class two. This may be because *Rhododendron* plants mostly reproduce asexually through basal germination [25], but radial growth is slow. Additionally, the natural regeneration of seeds in *Rhododendron* plants requires extremely strict habitat conditions, and site conditions often limit their seed germination and seedling survival [26]. The survival rate of seedlings at age class one is lower than that at age class two, resulting in far lower individual numbers at age class one than at age class two.

The two *Rhododendron* populations in communities with different orders of succession were both expanding populations; however, they differed in age and height structures. The age structures of *R. annae* in communities I_a_ and II_a_ presented an “inverted-J-shaped” pattern, whereas the age structure of *R. annae* in community III_a_ and the age structures of *R. irroratum* in all three communities showed an irregular pyramid pattern. Similarly, the height structures of *R. annae* in the three communities and those of *R. irroratum* in communities I_b_ and III_b_ showed an irregular pyramid pattern, whereas the height structure of *R. irroratum* in community II_b_ presented a “J-shaped” pattern. Because of differences in micro-environmental factors (such as light, nutrients, and survival space) between the two communities with different orders of succession, varying competition intensity within and between species further resulted in differences in population size. As a result, certain differences were found in age and height structures between the two *Rhododendron* populations. Both *Rhododendron* populations in communities with different orders of succession—characterized by large numbers of saplings, high survival rate at low age classes, and low survival rate at middle and high age classes—were classified as expanding populations. This classification was similar to those of Yang et al. [27]. Perhaps because young individuals only require few environmental resources for growth and development and only face mild interspecific and intraspecific competition, the presence of a large number of young individuals can be maintained. With increasing tree age, the individual numbers of the two *Rhododendron* populations decrease rapidly, mainly because of resource limitations as well as self-thinning and allelopathic effects [26,28,29]. Faced with intensified interspecific and intraspecific competition, the individual numbers of both populations experience a gradual decline. This observation explains why both *Rhododendron* populations had many young and middle-aged individuals but few old individuals. To sum up, the proportion of seedlings and young trees in two *Rhododendron* populations was large, and although the mortality rate of young individuals was relatively high, the period was still critical and sensitive. Therefore, further efforts should be made to protect *Rhododendron* shrubs and scientific measures should be taken to avoid human interference and prevent population reduction.

### 3.2. Population Dynamics and Development

The quantitative dynamics of a population reflect the interaction between its individual survivability and the environment, which is substantially influenced by the external environment. Time sequence analysis can predict the dynamics of plant populations to a certain extent [30]. This study showed that the quantitative dynamic indices of both *Rhododendron* populations with different orders of succession (*V*_pi_ and *V′*_pi_) were both greater than 0. The *V′*_pi_ of *R. annae* in community II_a_ was much greater than that in communities I_a_ and III_a_, and the *V′*_pi_ of *R. irroratum* in communities II_b_ and III_b_ was much greater than that in community I_b_. This clarified that both *Rhododendron* populations held dominant positions and grew rapidly in the two *Rhododendron* communities with different orders of succession. The *V′*_pi_ of *R. annae* was much greater than that of *R. irroratum*, and *R. annae* had few surviving individuals in both age classes five and six. This result suggests that although *R. annae* and *R. irroratum* are both expanding populations, *R. irroratum* grows slower and is more sensitive to the external environment. The survival, mortality rate, and disappearance rate curves of a population can be used to analyze its quantitative dynamics and gauge its development trends, thereby explaining the interactions between plant populations and the environment [6,7]. As indicated by the static life tables of the populations of *R. annae* and *R. irroratum*, with increasing age class, the mortality and disappearance rates of both populations gradually increased. This may be because, as they age, *Rhododendron* populations experience increasing demands for resources. In particular, *Rhododendron* individuals entering the overstory face intensified competition in terms of nutrients, light, moisture, and other resources with increasing crown breadth. By contrast, old *Rhododendron* individuals are in the stage of physiological aging, which encompasses a sharp weakening of competitiveness and the occurrence of mass mortality.

The largest quantities of *R. annae* were observed at age class six in communities I_a_ and II_a_ and at age class three in community III_a_; the largest quantities of *R. irroratum* were observed at age class six in community I_b_ and at age class three in communities I_b_ and II_b_. This result indicated that the mortality and disappearance rates of both populations gradually increased over their succession and peaked at age class three. Because the largest quantity of *Rhododendron* individuals was found at age class three, a self-thinning effect resulted under the combined action of density constraints and competition. At higher age classes, the quantity declined because of physiological decline. The life expectancies and survival rates of *R. annae* and *R. irroratum* were consistent with their mortality and disappearance rate trends. In the two orders of succession, the life expectancies of the *Rhododendron* populations at age classes one and two were significantly higher than those at other age classes. To reduce intraspecific competition among *Rhododendron* seedlings, relieve the self-thinning of seedlings and saplings caused by density constraints, and improve the efficiency of population supplementation from low to high age classes [31], the recommendation is to appropriately reduce the seedling numbers of both *Rhododendron* populations, thereby raising the early survival rate of *Rhododendron* individuals. Both quantities of *R. annae* and *R. irroratum* showed a trend of gradual increase after two, four, and six years at each age class, indicating that both *Rhododendron* populations were expanding populations. The survival curve test showed that both *Rhododendron* populations belonged to the Deevey-II type in communities with different orders of succession and were both stable populations. Although *R. annae* and *R. irroratum* had large population sizes in communities II_a_, III_a_, and II_b_, their individual numbers were small in communities I_a_ and I_b_, especially at high age classes. In this context, if the survival rate of younger individuals cannot be improved, the old individuals in these communities will not be effectively supplemented, thus potentially causing the population to decline or even disappear [27].

The populations of *R. annae* and *R. irroratum* in northwest Guizhou have rich reserves of seedlings and saplings; therefore, they are expanding populations. Without human interference, *Rhododendron* populations can naturally reproduce and realize self-renewal and rejuvenation. However, considering the poor environmental adaptability and low survival rate of young *Rhododendron* individuals, protective measures should be implemented. The goal of these measures should be to increase both the quantity and survival rate of *Rhododendron* seedlings and saplings and promote the natural renewal of *Rhododendron* populations. At the same time, the on-site protection of existing *Rhododendron* populations should be strengthened to reduce human interference and protect plants from further damage. To effectively protect the germplasm resources of *Rhododendron*, efforts should also be made to conduct breeding, cultivation, and management of *Rhododendron* populations. Moreover, breeding and protection bases for them should be established, their quantities should be increased, and their spatial distribution should be expanded.

## 4. Materials and Methods

### 4.1. Overview of the Study Area

The study area is located in Xingxiu Township, Dafang County, Guizhou Province, China (27°23′ N, 105°51′ E), which is a transition zone from the highest plateau surface to the central plateau surface in northwest Guizhou. Its terrain is low in the north and south and high in the middle (Figure 5). It has a typical karst peak-cluster mid-slotted trough valley topography, with an altitude of 1730–1820 m, and a subtropical monsoon humid climate. The annual average temperature is 11.8 °C, and the average annual precipitation is 1150.4 mm. The frost-free period is 257 d, and the annual sunshine duration is 1335.5 h. The zonal vegetation is a mountain evergreen broad-leaved forest, and the existing vegetation is dominated by *Rhododendron* shrubs, with important successional and transitional characteristics [32,33]. The major dominant species include *R. annae*, *R. irroratum*, *R. maculiferum*, *R. lilacinum*, *R. maculatum*, and *Lyonia ovalifolia*. Most of the dominant species belong to the *Rhododendron* genus including shrubs or trees. Their leaves are evergreen or deciduous, semi deciduous, alternate, entire, rare and have inconspicuous small teeth. Their flower buds are mostly characterized by bud scales with varying shapes and sizes. The flowers are prominent, small to large in shape, and are usually arranged in umbrella-shaped or short racemes, with sparse single flowers, which are usually terminal and rarely axillary, corolla funnel shaped, bell shaped, tubular, or high-footed disc shaped, neat or slightly symmetrical, with lobes covered in tiles within the bud. The appearance of two types of *Rhododendron* is shown in Figure 6. The dominant soil type is yellow soil, with a pH value of 4.61–5.32 [22].

### 4.2. Sample Plot Setting and Survey Methods

In March 2019, a survey was conducted on two orders of succession of the communities of *R. annae* and *R. irroratum* in the Baili Azalea Nature Reserve. To be specific, the orders of succession of *R. annae* included the three successional communities of *R. annae* + *Lyonia ovalifolia* shrubs (I_a_), *R. annae* shrubs (II_a_), and *R. annae* + evergreen forest (III_a_). The successional communities of *R. irroratum* also included three successional communities, namely, *R. irroratum* + *Lyonia ovalifolia* shrubs (I_b_), *R. irroratum* shrubs (II_b_), and *R. irroratum* + evergreen forest (III_b_) (Figure 5). Both orders of succession showed an alternation from southeast to northwest. The basic characteristics of all six communities are provided in Table 5. The constructive species of the *R. annae* succession changed from *Lyonia ovalifolia* to *R. annae*, and *R. irroratum* succession changed from *Lyonia ovalifolia* to *R. annae*.

A 20 m × 20 m quadrat was set up in each of the six *Rhododendron* shrub communities mentioned above, with six such quadrats in total. The longitude, latitude, and altitude of the center point of each quadrat were recorded. In April, July, and August 2019, the types, stem base diameters, heights, crown breadths, and cluster numbers of all shrubs with a stem base diameter (trunk diameter 5 cm above ground) of >1 cm in each quadrat were examined. 

### 4.3. Research Indicators and Data Analysis

#### 4.3.1. Quantification of Population Dynamics

The analysis method of replacing age structure with diameter class structure [34] was adopted to divide *R. annae* and *R. irroratum* in the above quadrats into six age classes and six height classes according to their stem base diameter and height, respectively. To be specific, the age classes included age class i of <2 cm (ii, 2–4 cm; iii, 4–6 cm; iv, 6–8 cm; v, 8–10 cm; vi, >10 cm). The height classes included height class i of <1 m (ii, 1–2 m; iii, 2–3 m; iv, 3–4 m; v, 4–5 m; vi, >6 m).

A quantitative method was employed to analyze the dynamics of the individual number between adjacent diameter classes for the populations of *R. annae* and *R. irroratum* in the two *Rhododendron* communities with different orders of succession [35]. The formula is as follows:(1)$$V_{n}=\frac{S_{n}-S_{n+1}}{\max\left(S_{n},S_{n+1}\right)}\times 100\%$$
(2)$$V_{pi}=\frac{1}{\sum_{n=1}^{k-1}S_{n}}\times \sum_{n=1}^{k-1}\left(S_{n}{\times V}_{n}\right)$$
where *V_n_* is the dynamics of individual number for a population from diameter class *n* to *n* + 1; *V_pi_* is the quantitative dynamic index of the entire population structure; *k* is the diameter class number of the population; *S_n_* and *S_n_*_+1_ are the individual numbers of the population at diameter classes *n* and *n* + 1, respectively. When external interference is considered,
(3)$${V^{\prime }}_{pi}=\frac{1\sum_{n=1}^{k-1}\left(S_{n}\times V_{n}\right)}{k\times \min\left(S_{1},S_{2},S_{3},\ldots ,S_{k}\right)\times \sum_{n=1}^{k-1}S_{n}}.$$

The positive, negative, and zero values of *V_pi_* and *V_n_* reflect the growth, decline, and stability of the individual number for the population or between adjacent age classes.

#### 4.3.2. Static Life Tables and Survival Curves

Static life tables were used to analyze the dynamic changes among both *Rhododendron* populations [3,36]. The “smoothing technique” was applied for data processing. To create static life tables, *a_x_* was replaced by *a′_x_* [37]. A static life table includes the following parameters: *a_x_*: existing individual number within age class *x*; *a′_x_*: existing individual number within age class *x* after the application of the “smoothing technique”; *l_x_*: standardized survival number at the beginning of age class *x* (generally converted to 1000); *lnl_x_*: logarithmic standardized survival number; *d_x_*: standardized death number within the interval from age class *x* to *x* + 1; *q_x_*: mortality rate; *L_x_*: average survival number within the interval from age class *x* to *x* + 1; *T_x_*: total survival number from age class *x* and beyond; *e_x_*: life expectancy of individuals entering age class *x*; *S_x_*: survival rate; *K_x_*: disappearance rate of the population. The formulae are as follows:(4)$$l_{x}=\frac{a_{x}}{a_{0}}\times 1000$$
(5)$$q_{x}=\frac{d_{x}}{l_{x}}$$
(6)$$L_{x}=\frac{l_{x}+l_{x+1}}{2}$$
(7)$$T_{x}=\sum L_{x}$$
(8)$$e_{x}=\frac{T_{x}}{l_{x}}$$
(9)$$S_{x}=\frac{l_{x+1}}{l_{x}}$$
(10)$$K_{x}={lnl}_{x}-{lnl}_{x+1}$$

To test whether the survival status of a population conforms to a Deevey-type II or Deevey-type III curve, this study adopted an exponential equation (y = *a* · *e^bx^*, where a and b are constants) and a power function equation (y = *a* · *x^b^*, where a and b are constants) to test the survival curves of the populations of *R. annae* and *R. irroratum* in communities at different stages of succession [38].

#### 4.3.3. Time Sequence Prediction

The renewal ability of *Rhododendron* populations was simulated and predicted using the moving average method [39]. The formula is as follows:$$M_{t}^{\left(1\right)}=\frac{1}{n}\times \sum_{k=t-n+1}^{t}x_{k}$$
where *n* is the future period to be predicted; *M_t_*^(1)^ is the size of the population at diameter class *t* in the future *n* years; *x_k_* is the current size of the population at diameter class *k*.

### 4.4. Data Processing and Statistical Analysis

The data were processed using Microsoft Office Excel 2016. The diameter class/height structure differences, survival statuses, and time sequence predictions of populations in communities at different stages of succession were analyzed using SPSS 26.0 and SigmaPlot 14.0 software was used to draw plots.

## Figures and Tables

**Figure 1 plants-13-00946-f001:**
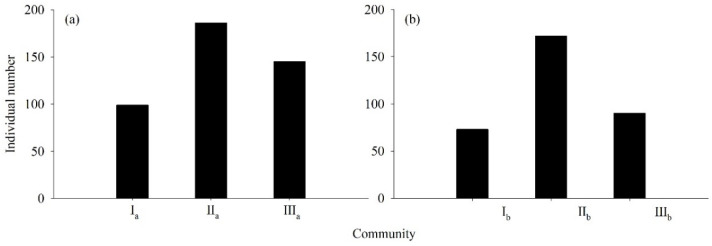
Quantities of *Rhododendron* plants in the quadrats. Note: (**a**), the individual quantity of *R*.*annae* in different communities; (**b**), the individual quantity of *R. irroratum* in different communities; I_a_, *R. annae* + *Lyonia ovalifolia* shrubs; II_a_, *R. annae* shrubs; III_a_, *R. annae* + broad-leaved forest; I_b_, *R. irroratum* + *Lyonia ovalifolia* shrubs; II_b_, *R. irroratum* shrubs; III_b_, *R. irroratum* + broad-leaved forest.

**Figure 2 plants-13-00946-f002:**
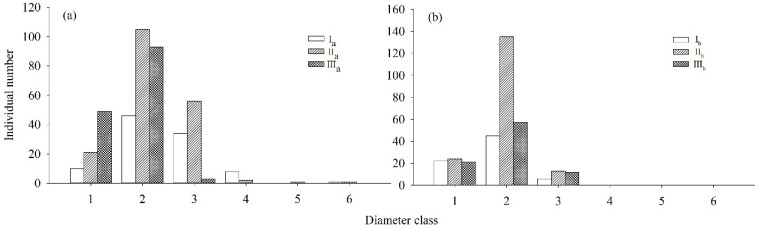
Age structures of *R. annae* and *R. irroratum* in different communities. Note: (**a**), age structures of *R. annae* in different communities; (**b**), age structures of *R. irroratum* in different communities. Note: I_a_, *R. annae* + *Lyonia ovalifolia* shrubs; II_a_, *R. annae* shrubs; III_a_, *R. annae* + broad-leaved forest; I_b_, *R. irroratum* + *Lyonia ovalifolia* shrubs; II_b_, *R. irroratum* shrubs; III_b_, *R. irroratum* + broad-leaved forest.

**Figure 3 plants-13-00946-f003:**
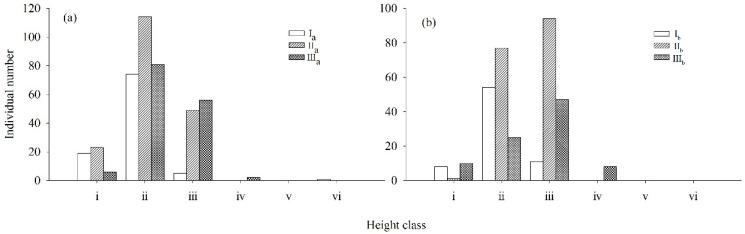
Height structures of *R. annae* and *R. irroratum* in different communities. Note: (**a**), height structures of *R. annae* in different communities; (**b**), height structures of *R. irroratum* in different communities. Note: I_a_, *R. annae* + *Lyonia ovalifolia* shrubs; II_a_, *R. annae* shrubs; III_a_, *R. annae* + broad-leaved forest; I_b_, *R. irroratum* + *Lyonia ovalifolia* shrubs; II_b_, *R. irroratum* shrubs; III_b_, *R. irroratum* + broad-leaved forest.

**Figure 4 plants-13-00946-f004:**
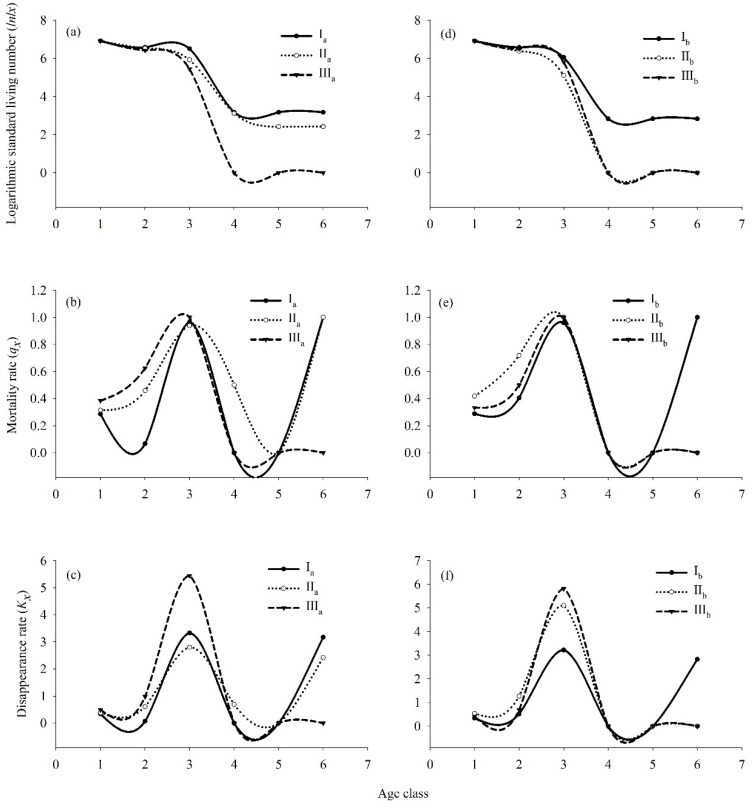
Survival, mortality rate (*q_x_*), and disappearance rate (*K_x_*) curves of *R. annae* and *R. irroratum* in different communities. Note: (**a**), survival curves of *R. annae* in different communities; (**b**), mortality rate curves of *R. annae* in different communities; (**c**), disappearance rate curves of *R. annae* in different communities; (**d**), survival curves of *R. irroratum* in different communities; (**e**), mortality rate of *R. irroratum* in different communities; mortality rate of *R. irroratum* in different communities; (**f**), disappearance rate of *R. irroratum* in different communities; I_a_, *R. annae* + *Lyonia ovalifolia* shrubs; II_a_, *R. annae* shrubs; III_a_, *R. annae* + broad-leaved forest; I_b_, *R. irroratum* + *Lyonia ovalifolia* shrubs; II_b_, *R. irroratum* shrubs; III_b_, *R. irroratum* + broad-leaved forest.

**Figure 5 plants-13-00946-f005:**
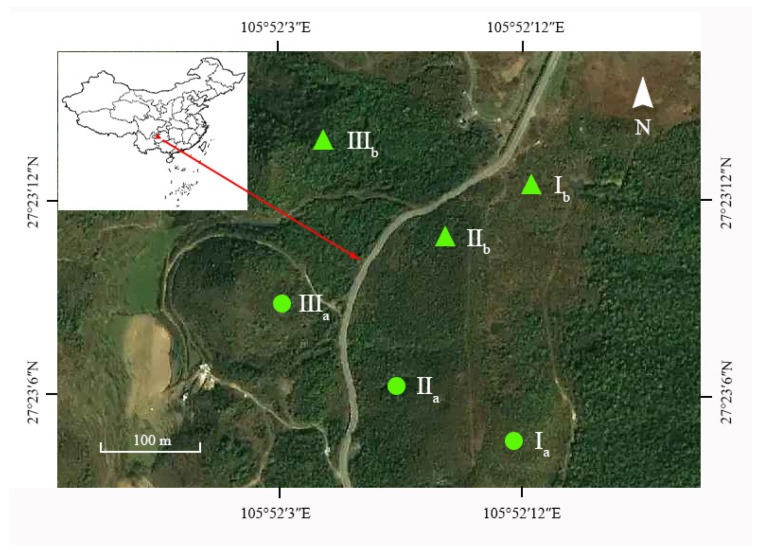
Location of the study area. Note: I_a_, *R. annae* + *Lyonia ovalifolia* shrubs; II_a_, *R. annae* shrubs; III_a_, *R. annae* + broad-leaved forest; I_b_, *R. irroratum* + *Lyonia ovalifolia* shrubs; II_b_, *R. irroratum* shrubs; III_b_, *R. irroratum* + broad-leaved forest.

**Figure 6 plants-13-00946-f006:**
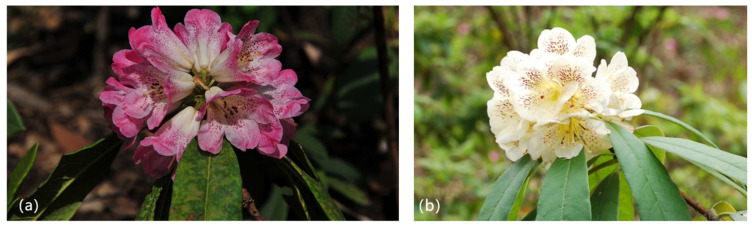
Photographs of *R. annae* (**a**) and *R. irroratum* (**b**).

**Table 1 plants-13-00946-t001:** Dynamic indices for the age structures of *Rhododendron* plants.

Index (%)	Community					
	I_a_	II_a_	III_a_	I_b_	II_b_	III_b_
*V* _1_	−78.26	80.00	−47.31	−51.11	−82.22	−63.16
*V* _2_	26.01	46.67	96.77	86.67	90.37	78.95
*V* _3_	76.47	96.43	100.00	100.00	100.00	100.00
*V* _4_	100.00	50.00	–	–	–	–
*V* _5_	−100.00	0.00	–	–	–	–
*V* _6_	100.00	100.00	–	–	–	–
*V_pi_*	39.57	47.42	48.15	46.24	67.02	48.60
*V′_pi_*	6.59	7.90	2.67	1.28	0.86	0.76

Note: I_a_, *R. annae* + *Lyonia ovalifolia* shrubs; II_a_, *R. annae* shrubs; III_a_, *R. annae* + broad-leaved forest; I_b_, *R. irroratum* + *Lyonia ovalifolia* shrubs; II_b_, *R. irroratum* shrubs; III_b_, *R. irroratum* + broad-leaved forest. *V_n_*: quantitative dynamic index between age classes *n* and *n* + 1; *V_pi_*: quantitative dynamic index of the population without considering external interferences; *V′_pi_*: quantitative dynamic index of the population considering external interferences.

**Table 2 plants-13-00946-t002:** Static life tables of *R. annae* and *R. irroratum* in different communities.

Species	Index	Community	Age Class					
			1	2	3	4	5	6
*R. annae*	*a_x_* (plant)	I_a_	10	46	34	8	0	1
		II_a_	21	105	56	2	1	1
		III_a_	49	93	3	0	0	0
	*a′_x_* (plant)	I_a_	42	30	28	1	1	1
		II_a_	89	61	33	2	1	1
		III_a_	78	48	18	0	0	0
	*l_x_* (plant)	I_a_	1000	714	667	24	24	24
		II_a_	1000	685	371	22	11	11
		III_a_	1000	615	231	–	–	–
	*lnl_x_*	I_a_	6.91	6.57	6.5	3.17	3.17	3.17
		II_a_	6.91	6.53	5.92	3.11	2.42	2.42
		III_a_	6.91	6.42	5.44	–	–	–
	*d_x_* (plant)	I_a_	286	48	643	0	0	24
		II_a_	315	315	348	11	0	11
		III_a_	385	385	231	–	–	–
	*q_x_* (%)	I_a_	28.57	6.67	96.43	0	0	100
		II_a_	31.46	45.9	93.94	50	0	100
		III_a_	38.46	62.5	100	–	–	–
	*L_x_* (plant)	I_a_	857	690	345	24	24	12
		II_a_	843	528	197	17	11	6
		III_a_	808	423	115	–	–	–
	*T_x_* (plant)	I_a_	1952	1095	405	60	36	12
		II_a_	1601	758	230	34	17	6
		III_a_	1346	538	115	–	–	–
	*e_x_* (year)	I_a_	1.95	1.53	0.61	2.5	1.5	0.5
		II_a_	1.6	1.11	0.62	1.5	1.5	0.5
		III_a_	1.35	0.88	0.5	–	–	–
	*K_x_* (%)	I_a_	0.34	0.07	3.33	0	0	3.17
		II_a_	0.38	0.61	2.8	0.69	0	2.42
		III_a_	0.49	0.98	5.44	–	–	–
*R. irroratum*	*a_x_* (plant)	I_b_	53	61	11	1	0	0
		II_b_	24	135	13	0	0	0
		III_b_	21	57	12	0	0	0
	*a′_x_* (plant)	I_b_	59	42	25	1	1	1
		II_b_	98	57	16	0	0	0
		III_b_	45	30	15	0	0	0
	*l_x_* (plant)	I_b_	1000	712	424	17	17	17
		II_b_	1000	582	163	–	–	–
		III_b_	1000	667	333	–	–	–
	*lnl_x_*	I_b_	6.91	6.57	6.05	2.83	2.83	2.83
		II_b_	6.91	6.37	5.1	–	–	–
		III_b_	6.91	6.5	5.81	–	–	–
	*d_x_* (plant)	I_b_	288	288	407	0	0	17
		II_b_	418	418	163	–	–	–
		III_b_	333	333	333	–	–	–
	*q_x_* (%)	I_b_	28.81	40.48	96	0	0	100
		II_b_	41.84	71.93	100	–	–	–
		III_b_	33.33	50	100	–	–	–
	*L_x_* (plant)	I_b_	856	568	220	17	17	8
		II_b_	791	372	82	–	–	–
		III_b_	833	500	167	–	–	–
	*T_x_* (plant)	I_b_	1686	831	263	42	25	8
		II_b_	1245	454	82	–	–	–
		III_b_	1500	667	167	–	–	–
	*e_x_* (year)	I_b_	1.69	1.17	0.62	2.5	1.5	0.5
		II_b_	1.24	0.78	0.5	–	–	–
		III_b_	1.5	1	0.5	–	–	–
	*K_x_* (%)	I_b_	0.34	0.52	3.22	0	0	2.83
		II_b_	0.54	1.27	5.1	–	–	–
		III_b_	0.41	0.69	5.81	–	–	–

Note: I_a_, *R. annae* + *Lyonia ovalifolia* shrubs; II_a_, *R. annae* shrubs; III_a_, *R. annae* + broad-leaved forest; I_b_, *R. irroratum* + *Lyonia ovalifolia* shrubs; II_b_, *R. irroratum* shrubs; III_b_, *R. irroratum* + broad-leaved forest. *a_x_*: plant number within age class x; *a′_x_*: plant number within age class x after smoothing; *l_x_*: standardized survival number; *lnl_x_*: logarithmic standardized survival number; *d_x_*: death number; *q_x_*: mortality rate; *L_x_*: survival number within the interval from age class x to x + 1; *T_x_*: total survival number; *e_x_*: life expectancy; *K_x_*: disappearance rate.

**Table 3 plants-13-00946-t003:** Function models for *R. annae* and *R. irroratum* in different communities.

Community	Exponential Function Model	*R^2^*	Power Function Model	*R^2^*
I_a_	*y* = 9.948*e*^−0.235*x*^	0.664 **	*y* = 7.831*x*^−0.438^	0.501 **
II_a_	*y* = 10.733*e*^−0.284*x*^	0.879 **	*y* = 8.445*x*^−0.640^	0.720 **
III_a_	*y* = 7.908*e*^−0.120*x*^	0.953 **	*y* = 7.040*x*^−0.206^	0.870 **
I_b_	*y* = 10.529*e*^−0.276*x*^	0.722 **	*y* = 7.980*x*^−0.520^	0.555 **
II_b_	*y* = 8.234*e*^−0.152*x*^	0.933 **	*y* = 7.096*x*^−0.259^	0.840 **
III_b_	*y* = 7.600*e*^−0.087*x*^	0.972 **	*y* = 6.991*x*^−0.150^	0.902 **

Note: “**” indicates a significant correlation at the 0.01 level. I_a_, *R. annae* + *Lyonia ovalifolia* shrubs; II_a_, *R. annae* shrubs; III_a_, *R. annae* + broad-leaved forest; I_b_, *R. irroratum* + *Lyonia ovalifolia* shrubs; II_b_, *R. irroratum* shrubs; III_b_, *R. irroratum* + broad-leaved forest.

**Table 4 plants-13-00946-t004:** Time sequence analysis of population dynamics of *R. annae* and *R. irroratum* in different communities.

Species	Age class	Primary Data			M_2_^(1)^			M_4_^(1)^			M_6_^(1)^		
		I_a_	II_a_	III_a_	I_a_	II_a_	III_a_	I_a_	II_a_	III_a_	I_a_	II_a_	III_a_
*R. annae*	1	10	21	49									
	2	46	105	93	28	63	71						
	3	34	56	3	40	81	48						
	4	8	2	0	21	29	2	30	58	40			
	5	0	1	0	4	2	0	22	37	17			
	6	1	1	0	1	1	0	9	11	0	15	27	14
	Total	99	186	145	94	176	121	61	106	57	15	27	14
**Species**	**Age class**	**Primary Data**			**M_2_^(1)^**			**M_4_^(1)^**			**M_6_^(1)^**		
		**I_b_**	**II_b_**	**III_b_**	**I_b_**	**II_b_**	**III_b_**	**I_b_**	**II_b_**	**III_b_**	**I_b_**	**II_b_**	**III_b_**
*R. irroratum*	1	53	24	21									
	2	61	135	57	57	80	39						
	3	11	13	12	36	74	35						
	4	1	0	0	6	7	6	33	54	27			
	5	0	0	0	1	0	0	14	27	14			
	6	0	0	0	0	0	0	2	2	0	12	21	10
	Total	126	172	90	100	161	80	49	83	41	12	21	10

Note: I_a_, *R. annae* + *Lyonia ovalifolia* shrubs; II_a_, *R. annae* shrubs; III_a_, *R. annae* + broad-leaved forest; I_b_, *R. irroratum* + *Lyonia ovalifolia* shrubs; II_b_, *R. irroratum* shrubs; III_b_, *R. irroratum* + broad-leaved forest. M_2_^(1)^, M_4_^(1)^, M_6_^(1)^ is a prediction of the population after the time of age class 2, 4 and 6, respectively.

**Table 5 plants-13-00946-t005:** Basic situation of different *Rhododendron* shrub communities.

Community Type	Mark	Altitude (m)	Longitude and Latitude	*Rhododendron* Important Value	Constructive Species
*R. annae* + *Lyonia ovalifolia* shrubs	I_a_	1807	E 105°51′03.52″ N 27°24′5.53″	0.453	*Lyonia ovalifolia* *R. annae*
*R. annae* shrubs	II_a_	1825	E 105°51′58.12″ N 27°23′20.93″	0.595	*R. annae*
*R. annae* + broad-leaved forest	III_a_	1783	E 105°51′52.23″ N 27°23′23.94″	0.485	*R. annae*
*R. irroratum* + *Lyonia ovalifolia* shrubs	I_b_	1803	E 105°52′58.88″ N 27°23′25.34″	0.417	*R. irroratum* *Lyonia ovalifolia*
*R. irroratum* shrubs	II_b_	1804	E 105°51′57.49″ N 27°23′24.64″	0.656	*R. irroratum*
*R. irroratum* + broad-leaved forest	III_b_	1811	E 105°51′52.78″ N 27°23′22.93″	0.556	*R. irroratum*

## Data Availability

The original contributions presented in the study are included in the article, further inquiries can be directed to the corresponding author.

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
