# Peer review of "Population Structures and Dynamics of Rhododendron Communities with Different Stages of Succession in Northwest Guizhou, China"

_plants, 2024, doi:10.3390/plants13070946_

Round 1

Reviewer 1 Report

Comments and Suggestions for Authors

Dear Author,

The manuscript "Population structures and dynamics of Rhododendron communities with different orders of succession in northwest Guizhou, China" is written in a comprehensible and clear language. The article is very interesting in terms of content and provides a lot of information about population ecology of two species of Rododendron genera in natural habitats.

Introduction provide a lot of worth and actual information and efficiently introduces the reader to the subject of the manuscript.  The Authors clearly presented the aims of the research and the rationale for undertaking them.  In my opinion, the Authors should briefly describe the species studied to show their unique character and compare their characteristic features. My suggestion is to show Rododendron annae and R. irroratum on photographs to familiarize readers with the appearance of plants.

The research methods are described precisely and clearly. It seems to me that it would be more readable to present the text first and then the cited figures – Fig. 1 (the same applies to the figures in the Results section – fig. 2, 3, 4 and table 2). Line 103 should include a citation to the source of the information contained in the last sentence of this paragraph.

The results were described in detail. I would suggest adding an introductory sentence in chapter 3.1. because the text starts with a description of the Fig. 2, which is not very elegant.

A large number of obtained results were discussed and compared with available literature sources. However, I personally miss a short summary of the most important conclusions in a separate section (perhaps the Authors should consider this option).

Apart from these minor comments, I have no other suggestions.

I hope that the materials collected by Authors will be published after minor revision.

Author Response

Dear Reviewer
We have responded to your comments one by one, please  see the attachment.

Reviewer 2 Report

Comments and Suggestions for Authors

Plants-2902878

Population structures and dynamics of Rhododendron Communities with Different Orders of Succession in Northwest Guizhou, China

General comments

This study is interesting about the population structure and dynamics of a dominant species in China. The study includes all parameters that should be evaluated in a population.  Also, the document is well-written and has most of the parts of a scientific paper. 

However, I have two major observations in the manuscript. The first observation refers to the objectives, which are not very visible in the document. It would be good if they could be placed in the last paragraph of the Introducction section. The second observation refers to the study design. I see that there is only one 20x20m plot per successional stage for each species studied. I don't know if that is enough for a study of this magnitude. It would be good if this part could be clarified. Finally, I have some minor observations, especially to the redundancy in the use of some terms and other additional aspects.

Specific comments

Lines / Sections

Comments

3

I think the best word instead of “Orders” is Stages

13, 14

In these two lines I see redundancy using “succession” term

13

I think there should be “populations” instead of “communities” 

59

Include the Family to which belong Rhododendron

87

You should write the goal and specific objectives of this study

91

Explain the symbols and colors on the map

92

According to the scale of the map, it seems a very small site in which you did your study

103-104

I think you need to include a section with a paragraph describing the main characteristics of Rhododendron genus.

Table 1

What is the source of Rhododendron important value? Constructive species, is similar to “dominant species”?

115

I think you need to complete, considering different characteristics that you have in this Table

116

You have only 6 quadrants of 20mx20m, and this is all you have? It seems to be not representative, considering the size of plants. I think this study needs replicates. Please justify why you have only one plot for each succession stage.

119-122 and 129-132

There are redundancy between these sections

180

I think you need to include “and” between SPSS 26.0 and Sigma Plot 14.0

213-222

I think you need to explain the biological meaning instead of explaining if it is less or greater than 0

258-259

Title is very large

273

Put text to the top line

Table 5

You need to explain better the content of the Table. For example, what means Primary data, M2, M4, M6?

482

 “populus” should be “Populus”

497,505

 The titles of references need to be written in consistent capital and small letter. Such as “different treatments” and “Macromorphological Characters”

507

“of13Species” should add the space

512

“Rhododendron” should be in italics

Author Response

(The authors gave the same response as above.)

Round 2

Reviewer 2 Report

Comments and Suggestions for Authors

Authors have worked hard to have a good version of their manuscript. I think it is ready to be accepted and published.

Author Response

I am very grateful to your comments for the manuscript. Wishing you a happy life and smooth work.